# Interaction effects between insomnia and depression on risk of out-of-hospital cardiac arrest: Multi-center study

**Eujene Jung**[1], **Hyun Ho Ryu**[1,2]*, **Sung Wan Kim**[3], **Jung Ho Lee**[1], **Kyoung Jun Song**[4], **Young Sun Ro**[5], **Kyoung Chul Cha**[6], **Sung Oh Hwang**[6], **Phase II Cardiac Arrest Pursuit Trial with Unique Registry and Epidemiologic Surveillance (CAPTURES-II investigators)**[¶]

1 Department of Emergency Medicine, Chonnam National University Hospital, Gwangju, Korea, 2 Medicine, Chonnam National University, Gwangju, Korea, 3 Department of Psychiartry, Chonnam National University Medical School, Gwangju, Korea, 4 Department of Emergency Medicine, Seoul National University Boramae Medical Center, Seoul, Korea, 5 Department of Emergency Medicine, Seoul National University Hospital, Seoul, Korea, 6 Department of Emergency Medicine, Yonsei University Wonju College of Medicine, Wonju, Korea

¶ Membership of the Phase II Cardiac Arrest Pursuit Trial with Unique Registry and Epidemiologic Surveillance (CAPTURES-II investigators) is listed in the Acknowledgments.
* em.ryu.hyunho@gmail.com

**Data Availability Statement:** Data were obtained from the Korea Disease Control and Prevention Agency. (jhjeong1107@korea.kr)

## Abstract

### Background

Insomnia and depression have been known to be risk factors of several diseases, including coronary heart disease. We hypothesized that insomnia affects the out-of-hospital cardiac arrest (OHCA) incidence, and these effects may vary depending on whether it is accompanied by depression. This study aimed to determine the association between insomnia and OHCA incidence and whether the effect of insomnia is influenced by depression.

### Methods

This prospective multicenter case-control study was performed using Phase II Cardiac Arrest Pursuit Trial with Unique Registration and Epidemiology Surveillance (CAPTURES-II) project database for OHCA cases and community-based controls in Korea. The main exposure was history of insomnia. We conducted conditional logistic regression analysis to estimate the effect of insomnia on the risk of OHCA incidence and performed interaction analysis between insomnia and depression. Finally, subgroup analysis was conducted in the patients with insomnia.

### Results

Insomnia was not associated with increased OHCA risk (0.95 [0.64–1.40]). In the interaction analysis, insomnia interacted with depression on OHCA incidence in the young population. Insomnia was associated with significantly higher odds of OHCA incidence (3.65 [1.29–10.33]) in patients with depression than in those without depression (0.84 [0.59–1.17]). In the subgroup analysis, depression increased OHCA incidence only in patients who were not taking insomnia medication (3.66 [1.15–11.66]).

**Funding:** This work was supported by the Korea Disease Control and Prevention Agency (Grant No: 2017NE3300600, 2017E3300601, 2019P330800). There was no additional external funding received for this study. The funders had no role in study design, data collection and analysis, decision to publish, or preparation of the manuscript.

**Competing interests:** The authors have declared that no competing interests exist.

## Conclusion

Insomnia with depression is a risk factor for OHCA in the young population. This trend was maintained only in the population not consuming insomnia medication. Early and active medical intervention for patients with insomnia may contribute to lowering the risk of OHCA.

## Introduction

Sudden cardiac death (SCD) is a major public health burden because of its high incidence and low survival rates. The average global incidence of SCD among adults is approximately 55 per 100,000 person-years and 97.1 per 100,000 population in USA [1]. Survival to discharge rate is lower in Asia (4.5%) than that in North America (7.7%) and Europe (11.7%) [2]. Despite advances in resuscitation technology and post-resuscitation care for SCD patients, survival outcomes after SCD remain poor [3–5].

The approach to SCD risk stratification is difficult. Recent studies suggest that there are several unrecognized variables that directly affect cardiac function and the occurrence of arrhythmias related to the risk of SCD; however, the risk of SCD is also associated with classical factors, such as hypertension, diabetes mellitus, and dyslipidemia [6, 7].

Insomnia, the most common sleep disorder, is characterized by difficulty initiating and maintaining sleep. It may also take the form of early-morning awakening in which the individual awakens several hours early and is unable to resume sleeping [8]. The duration and continuity of sleep is associated with homeostasis of cardiovascular, metabolic, and immune systems, and insomnia has been shown to adversely influence metabolism and endocrine function, including altering the hypothalamic-pituitary-adrenal axis and elevating biomarkers of chronic inflammation [9–11]. Numerous previous studies have shown that insomnia is associated with an increased coronary heart disease and that it influences cardiovascular mortality; however, results to date have been inconsistent [12–14]

Depression, which is often accompanied with insomnia, not only increases the risk of a new event of heart disease but also increases the risk of recurrent events in patients with heart disease through several biological pathways [15, 16].

Insomnia and depression are commonly comorbid and are associated with significant distress, daytime fatigue, and increased likelihood of day-time sleep, in addition to pathological effects [17, 18]. Consequently, it can impair an individual's ability to adapt and respond to illness [19, 20].

Therefore, it is important to understand whether insomnia and depression may by associated with OHCA risk both independently and in conjunction. We hypothesized that insomnia affects the risk of OHCA through several processes and that these effects may vary depending on whether or not depression is present. The purpose of this study, therefore, was to determine the association of insomnia with the risk of OHCA and identify if its effect is influenced by depression.

## Methods

### Study design, setting, and data source

This study was performed using Phase II Cardiac Arrest Pursuit Trial with Unique Registration and Epidemiologic Surveillance (CAPTURES-II) project database in Korea.

The CAPTURES project is a prospective, multi-center, case-control study, which aims to identify risk factors for OHCA and evaluate prognostic factors with long-term follow-up. Phase I CAPTURES was performed in 27 hospitals from January to December 2014. A detailed description of Phase I CAPTURES has been previously reported [21].

The CAPTURES-II project was a prospective case-control study conducted at 17 university hospitals from September 2017 to December 2020 [21, 22]. The CAPTURES-II was a hospital-based open-cohort study, which included patients who suffered OHCA with a presumed cardiac etiology, were transported to the participating emergency department (ED) by emergency medical services (EMS) and identified by the emergency physicians, and matched community-based controls. The CAPTURES-II registry included face-to-face interview for demographics, comorbidities assessment, health behaviors monitoring, medical record review including laboratory test results, short- and long-term follow-ups (1 month, 6 months, and 12 months after hospital discharge), and blood sample collection for biomarker evaluation. For community-based controls, face-to-face interview and medical record review were conducted to obtain the same information concerning OHCA, and blood sample collection was performed. The data collected from all participating hospitals were transferred to the Quality Management Committee (QMC), where quality control and statistical analysis were performed. The QMC provided feedback to research coordinators on data quality control through monthly meetings. If the research coordinator was unable to define a coding element, they consulted the emergency physicians in the QMC for clarification.

The study was approved by the ethics committees in all participating institutions. In compliance with the Declaration of Helsinki, all participants, or their proxies, provided a written informed consent prior to their participation in the study. This study is registered at Clinical-Trials.gov (NCT03700203).

## Study population

Cases were defined as adult (aged over 18 years) EMS-treated OHCA patients who had a presumed cardiac etiology and visited participating hospitals from September 2017 to December 2020.

CAPTURES II control recruitment was conducted by two participating hospitals (one metropolitan and one non-metropolitan) by cooperating with various community centers and public health centers. Community-based controls who were voluntary applications were recruited as control group. The controls were randomly matched within the strata, including age (10-year intervals), sex, and type of residence (metropolitan vs. urban or rural), with a 1:1 ratio of controls to cases.

## Variables and measurements

The main exposure was history of insomnia as measured via face-to-face interviews with the patients and community-based controls or their families (for cases). Insomnia history was defined as a positive case of answering "yes" to the question "Have you ever been diagnosed with insomnia?" Depression history was defined in the same way.

The CAPTURES-II registry used the same questionnaire for cases and controls. We collected information on patient demographics (age, sex, residence area [metropolis vs. urban or rural], insurance, education level), comorbidities (hypertension, diabetes, dyslipidemia, stroke), health behaviors (smoking, alcohol, coffee, physical activity), and sleep-related variables, as well as obesity, work type, sleeping medication, snoring, and sleep apnea. Whether or not comorbidities and sleep-related history were diagnosed by a doctor was investigated. Alcohol drinking was defined as heavy alcohol drinking when men drank more than 15 drinks per

week and women drank more than 8 drinks per week and physical activity was defined as vigorous when exercising to the extent of sweating at least once a week, and moderate when exercising less than once a week [23, 24].

## Statistical analysis

Demographic findings of the OHCA case group and community-based control group were described. Categorical variables were analyzed using the chi-square test, and continuous variables with normal distribution were analyzed using the t-test. Case-control matching was performed based on age, sex, and residence (metropolis vs. urban or rural). For the case-control dataset, conditional logistic regression analysis was conducted to estimate the effect of insomnia and depression on the risk of OHCA incidence and to calculate the adjusted odds ratios (AORs) and 95% confidence intervals (CIs) after adjusting for potential confounders identified in directed acyclic graph models. We also conducted interaction analysis between history of insomnia and history of depression according to age group. Finally, subgroup analysis was performed in the patients with insomnia. In subgroup analysis, we conducted interaction analysis between history of depression and insomnia medication. All variables in the final model were assessed for multicollinearity, which was not detected in this analysis.

All statistical analyses were conducted using SAS version 9.4 (SAS Institute Inc., Cary, NC, USA). All p-values were two tailed, and p<0.05 was considered statistically significant.

## Ethics statements

This study was approved by the Institutional Review Board (IRB) of Chonnam National University Hospital (IRB No. 2017–285) and the requirement for informed consent was waived due to the retrospective nature of this study. This study followed the Strengthening the Reporting of Observational Studies in Epidemiology (STROBE) guidelines.

## Results

### Main results

A total of 1014 OHCA cases and 1014 community-based controls were included in this analysis.

The characteristics of the OHCA cases and community-based controls are shown in Table 1. And, the characteristics according to the history of insomnia and depression are reported Tables 2 and 3.

The results of conditional logistic regression models, including AORs (95% CIs), for OHCA with history of insomnia and depression are shown in Table 4. Insomnia was not associated with increased OHCA risk (AOR [95% CI]: 0.95 [0.64–1.40]). Depression also was not a risk factor of OHCA (0.86 [0.67–1.10]).

### Interaction analysis

We added an interaction term between history of insomnia and depression to the fully adjusted model. The AORs assessing the statistical interaction on OHCA risk from the conditional logistic regression analysis are shown in Table 5. Insomnia interacted with depression on OHCA incidence in the young population (0.84 [0.59–1.17] for patients without depression, 3.65 [1.29–10.33] for patients with depression, p for interaction <0.01). In subgroup analysis, history of depression was associated with significantly higher odds of OHCA incidence for patients taking insomnia medication (3.66 [1.15–11.66]) than those not taking insomnia medication (1.05 [0.99–12.22]) (p for interaction <0.01).

**Table 1. Characteristics of the out-of-hospital cardiac arrest case group and age-, sex-, and urbanization level-matched control group.**

| Variables | All | OHCA case | Community control | |
|---|---|---|---|---|
| | N (%) | N (%) | N (%) | p-value |
| All | 2,028 (100.0) | 1,014 (100.0) | 1,014 (100.0) | |
| Age | | | | 1.00 |
| 18–29 | 38 (1.9) | 19 (1.9) | 19 (1.9) | |
| 30–39 | 114 (5.6) | 57 (5.6) | 57 (5.6) | |
| 40–49 | 276 (13.6) | 138 (13.6) | 138 (13.6) | |
| 50–59 | 512 (25.2) | 256 (25.2) | 256 (25.2) | |
| 60–69 | 544 (26.8) | 272 (26.8) | 272 (26.8) | |
| 70–120 | 544 (26.8) | 272 (26.8) | 272 (26.8) | |
| Gender, female | 554 (27.3) | 277 (27.3) | 277 (27.3) | 1.00 |
| Metropolis, yes | 1,028 (50.7) | 514 (50.7) | 514 (50.7) | 1.00 |
| Comorbidity | | | | |
| Hypertension | 823 (40.6) | 454 (44.8) | 369 (36.4) | <0.01 |
| Diabetes mellitus | 411 (20.3) | 274 (27.0) | 137 (13.5) | <0.01 |
| Dyslipidemia | 365 (18.0) | 130 (12.8) | 235 (23.2) | <0.01 |
| Stroke | 115 (5.7) | 85 (8.4) | 30 (3.0) | <0.01 |
| Insurance, medical aid | 210 (10.4) | 110 (10.8) | 100 (9.9) | 0.47 |
| Education level, high | 646 (31.9) | 253 (25.0) | 393 (38.8) | <0.01 |
| Obesity, yes | 146 (7.2) | 35 (3.5) | 111 (10.9) | <0.01 |
| Married, yes | 1,544 (76.1) | 705 (69.5) | 839 (82.7) | <0.01 |
| Work type | | | | |
| Night work | 362 (17.9) | 100 (9.9) | 262 (25.8) | <0.01 |
| Shift work | 178 (8.8) | 32 (3.2) | 146 (14.4) | <0.01 |
| Physical activity | | | | <0.01 |
| Vigorous | 365 (18.0) | 115 (11.3) | 250 (24.7) | |
| Moderate | 365 (18.0) | 115 (11.3) | 250 (24.7) | |
| No | 549 (27.1) | 159 (15.7) | 390 (38.5) | |
| Coffee, yes | 697 (34.4) | 202 (19.9) | 495 (48.8) | <0.01 |
| Smoking | | | | <0.01 |
| Current smoker | 536 (26.4) | 343 (33.8) | 193 (19.0) | |
| Ex-smoker | 597 (29.4) | 252 (24.9) | 345 (34.0) | |
| Non-smoker | 895 (44.1) | 419 (41.3) | 476 (46.9) | |
| Alcohol drinking | | | | <0.01 |
| Heavy | 617 (30.4) | 309 (30.5) | 308 (30.4) | |
| Moderate | 661 (32.6) | 246 (24.3) | 415 (40.9) | |
| No | 750 (37.0) | 459 (45.3) | 291 (28.7) | |
| Sleeping pill, yes | 95 (4.7) | 60 (5.9) | 35 (3.5) | <0.01 |
| Sleep disorder | | | | |
| Insomnia | 543 (26.8) | 272 (26.8) | 271 (26.7) | 0.96 |
| Snoring | 1,073 (52.9) | 493 (48.6) | 580 (57.2) | <0.01 |
| Sleep apnea | 350 (17.3) | 188 (18.5) | 162 (16.0) | 0.13 |
| Depression, Yes | 146 (7.2) | 74 (7.3) | 72 (7.1) | <0.01 |

OHCA, out-of-hospital cardiac arrest

**Table 2. Characteristics of the study population according to the insomnia.**

| Variables | All | Insomnia (+) | Insomnia (-) | |
|---|---|---|---|---|
| | N (%) | N (%) | N (%) | p-value |
| All | 2,028 (100.0) | 543 (100.0) | 1,485 (100.0) | |
| Case-control | | | | 0.17 |
| OHCA case | 1,014 (50.0) | 272 (50.1) | 742 (50.0) | |
| Community control | 1,014 (50.0) | 271 (49.9) | 743 (50.0) | |
| Age | | | | <0.01 |
| 18–29 | 38 (1.9) | 7 (1.3) | 31 (2.1) | |
| 30–39 | 114 (5.6) | 29 (5.3) | 85 (5.7) | |
| 40–49 | 276 (13.6) | 80 (14.7) | 196 (13.2) | |
| 50–59 | 512 (25.2) | 112 (20.6) | 400 (26.9) | |
| 60–69 | 544 (26.8) | 159 (29.3) | 385 (25.9) | |
| 70–120 | 544 (26.8) | 156 (28.7) | 388 (26.1) | |
| Gender, female | 554 (27.3) | 183 (33.7) | 371 (25.0) | <0.01 |
| Metropolis, yes | 1,028 (50.7) | 297 (54.7) | 731 (49.2) | 0.86 |
| Comorbidity | | | | |
| Hypertension | 823 (40.6) | 249 (45.9) | 574 (38.7) | <0.01 |
| Diabetes | 411 (20.3) | 140 (25.8) | 271 (18.2) | <0.01 |
| Dyslipidemia | 365 (18.0) | 126 (23.2) | 239 (16.1) | <0.01 |
| Stroke | 115 (5.7) | 45 (8.3) | 70 (4.7) | <0.01 |
| Insurance, medical aid | 210 (10.4) | 66 (12.2) | 144 (9.7) | 0.47 |
| Education level, high | 646 (31.9) | 162 (29.8) | 484 (32.6) | <0.01 |
| Obesity, yes | 146 (7.2) | 55 (10.1) | 91 (6.1) | <0.01 |
| Married, yes | 1,544 (76.1) | 415 (76.4) | 1,129 (76.0) | <0.01 |
| Work type | | | | |
| Night work | 362 (17.9) | 91 (16.8) | 271 (18.2) | <0.01 |
| Shift work | 178 (8.8) | 57 (10.5) | 121 (8.1) | <0.01 |
| Physical activity | | | | <0.01 |
| Vigorous | 365 (18.0) | 78 (14.4) | 287 (19.3) | |
| Moderate | 549 (27.1) | 141 (26.0) | 408 (27.5) | |
| No | 1,114 (54.9) | 324 (59.7) | 790 (53.2) | |
| Coffee, yes | 697 (34.4) | 157 (28.9) | 540 (36.4) | <0.01 |
| Smoking | | | | <0.01 |
| Current smoker | 536 (26.4) | 142 (26.2) | 394 (26.5) | |
| Ex-smoker | 597 (29.4) | 156 (28.7) | 441 (29.7) | |
| Non-smoker | 895 (44.1) | 245 (45.1) | 650 (43.8) | |
| Alcohol drinking | | | | <0.01 |
| Heavy | 617 (30.4) | 170 (31.3) | 447 (30.1) | |
| Moderate | 661 (32.6) | 162 (29.8) | 499 (33.6) | |
| No | 750 (37.0) | 211 (38.9) | 539 (36.3) | |
| Sleeping pill, yes | 95 (4.7) | 91 (16.8) | 4 (0.3) | <0.01 |
| Sleeping disorder | | | | |
| Snoring | 1,073 (52.9) | 339 (62.4) | 734 (49.4) | <0.01 |
| Sleep apnea | 350 (17.3) | 119 (21.9) | 231 (15.6) | 0.13 |
| Depression, yes | 146 (7.2) | 77 (14.2) | 69 (4.6) | <0.01 |

OHCA, out-of-hospital cardiac arrest

**Table 3. Characteristics of the study population according to the depression.**

| Variables | All | Depression (+) | Depression (-) | |
|---|---|---|---|---|
| | N (%) | N (%) | N (%) | p-value |
| All | 1,748 (100.0) | 146 (100.0) | 1,602 (100.0) | |
| Case-control | | | | <0.01 |
| OHCA case | 996 (57.0) | 74 (50.7) | 922 (57.6) | |
| Community control | 752 (43.0) | 72 (49.3) | 680 (42.4) | |
| Age | | | | 1.00 |
| 18–29 | 34 (1.9) | 1 (0.7) | 33 (2.1) | |
| 30–39 | 98 (5.6) | 11 (7.5) | 87 (5.4) | |
| 40–49 | 242 (13.8) | 35 (24.0) | 207 (12.9) | |
| 50–59 | 448 (25.6) | 19 (13.0) | 429 (26.8) | |
| 60–69 | 472 (27.0) | 37 (25.3) | 435 (27.2) | |
| 70- | 454 (26.0) | 43 (29.5) | 411 (25.7) | |
| Gender, female | 469 (26.8) | 51 (34.9) | 418 (26.1) | 1.00 |
| Metropolis, yes | 890 (50.9) | 101 (69.2) | 789 (49.3) | 0.86 |
| Comorbidity | | | | |
| Hypertension | 727 (41.6) | 57 (39.0) | 670 (41.8) | <0.01 |
| Diabetes | 342 (19.6) | 37 (25.3) | 305 (19.0) | <0.01 |
| Dyslipidemia | 339 (19.4) | 29 (19.9) | 310 (19.4) | <0.01 |
| Stroke | 97 (5.5) | 14 (9.6) | 83 (5.2) | <0.01 |
| Insurance, medical aid | 170 (9.7) | 30 (20.5) | 140 (8.7) | 0.47 |
| Education level, high | 591 (33.8) | 34 (23.3) | 557 (34.8) | <0.01 |
| Obesity, yes | 141 (8.1) | 17 (11.6) | 124 (7.7) | <0.01 |
| Married, yes | 1,407 (80.5) | 106 (72.6) | 1,301 (81.2) | <0.01 |
| Work type | | | | |
| Night work | 346 (19.8) | 20 (13.7) | 326 (20.3) | <0.01 |
| Shift work | 171 (9.8) | 11 (7.5) | 160 (10.0) | <0.01 |
| Physical activity | | | | <0.01 |
| Vigorous | 347 (19.9) | 20 (13.7) | 327 (20.4) | |
| Moderate | 510 (29.2) | 31 (21.2) | 479 (29.9) | |
| No | 891 (51.0) | 95 (65.1) | 796 (49.7) | |
| Coffee, yes | 669 (38.3) | 41 (28.1) | 628 (39.2) | <0.01 |
| Smoking | | | | <0.01 |
| Current smoker | 451 (25.8) | 41 (28.1) | 410 (25.6) | |
| Ex-smoker | 539 (30.8) | 42 (28.8) | 497 (31.0) | |
| Non-smoker | 758 (43.4) | 63 (43.2) | 695 (43.4) | |
| Alcohol drinking | | | | <0.01 |
| Heavy | 545 (31.2) | 42 (28.8) | 503 (31.4) | |
| Moderate | 597 (34.2) | 47 (32.2) | 550 (34.3) | |
| No | 606 (34.7) | 57 (39.0) | 549 (34.3) | |
| Sleeping pill, yes | 86 (4.9) | 30 (20.5) | 56 (3.5) | <0.01 |
| Sleep disorder | | | | |
| Insomnia | 486 (27.8) | 77 (52.7) | 409 (25.5) | <0.01 |
| Snoring | 989 (56.6) | 79 (54.1) | 910 (56.8) | <0.01 |
| Sleep apnea | 328 (18.8) | 32 (21.9) | 296 (18.5) | 0.13 |

OHCA, out-of-hospital cardiac arrest

**Table 4. Multivariable conditional logistic regression analysis of insomnia and depression for out-of-hospital cardiac arrest.**

| OHCA incidence | OHCA cases/Community controls | Model 1 | | | Model 2 | | | Model 3 | | | Model 3 (subgroup) | | |
|---|---|---|---|---|---|---|---|---|---|---|---|---|---|
| | (n/n) | AOR | 95% CI | | AOR | 95% CI | | AOR | 95% CI | | AOR | 95% CI | |
| Insomina (-) | 742/743 | 1.00 | | | 1.00 | | | 1.00 | | | | | |
| Insomnia (+) | 272/271 | 0.92 | 0.74 | 1.13 | 1.00 | 0.80 | 1.25 | 0.95 | 0.64 | 1.40 | | | |
| Depression (-) | 900/937 | 1.00 | | | 1.00 | | | 1.00 | | | 1.00 | | |
| Depression (+) | 114/77 | 1.49 | 1.08 | 2.06 | 1.23 | 0.87 | 1.73 | 0.86 | 0.67 | 1.10 | 2.30 | 1.26 | 4.20 |
| Younger age (18–64) | 602/599 | 1.00 | | | 1.00 | | | 1.00 | | | 1.00 | | |
| Older age (65–120) | 412/415 | 1.07 | 0.75 | 1.53 | 1.18 | 0.81 | 1.70 | 1.40 | 0.93 | 2.12 | 1.19 | 0.57 | 2.48 |

OHCA, out-of-hospital cardiac arrest

## Discussion

The results of this multicenter case-control study demonstrated that insomnia and depression are not associated with an increased risk of OHCA. However, in the interaction analysis of insomnia and depression, insomnia with depression increased the risk of OHCA after adjusting for demographic characteristics and lifestyle behaviors, and this trend was significant only in the younger age group without insomnia medication. This research contributes to understanding the complex effects of insomnia and depression related to OHCA risk and will help develop strategies to reduce OHCA in the general population.

Sleep disturbance, including insomnia, has been reported as a risk factor of coronary heart disease and mortality in previous studies [25–27]. Insomnia has been shown to adversely influence physiological metabolism and endocrine function similar to that in premature aging, including altering the hypothalamic-pituitary-adrenal axis, reducing endogenous testosterone levels, and elevating biomarkers of chronic inflammation [9, 10, 28]. Contrary to the results of previous studies, insomnia alone did not increase OHCA risk in our study.

Previous studies suggested that depression and depressive symptoms are associated with an increased risk of coronary heart disease and OHCA [16, 29, 30]. This may be due to the fact that depression is associated with atherosclerosis, which is buildup of plaque in the arteries that can lead to cardiac events, alteration of the cardiac autonomic response, and a decrease in heart rate variability [31, 32]. Additionally, low red blood cell membrane levels of omega-3 polyunsaturated fatty acids, which is associated with an increased risk of sudden cardiac death, have been reported in patients with depressive disorders [33]. Omega-3 fatty acids have

**Table 5. Interaction analysis for OHCA incidence of insomnia according to depression.**

| OHCA incidence | Insomnia (-) | Insomnia (+) | | | p for interaction |
|---|---|---|---|---|---|
| | AOR | AOR | 95% CI | | |
| Whole population (18–120) | | | | | 0.03 |
| Depression (-) | 1.00 | 0.89 | 0.70 | 1.13 | |
| Depression (+) | 1.00 | 2.33 | 1.17 | 4.64 | |
| Young population (18–64) | | | | | <0.01 |
| Depression (-) | 1.00 | 0.84 | 0.59 | 1.17 | |
| Depression (+) | 1.00 | 3.65 | 1.29 | 10.33 | |
| Old population (65–120) | | | | | 0.17 |
| Depression (-) | 1.00 | 0.87 | 0.60 | 1.27 | |
| Depression (+) | 1.00 | 1.73 | 0.64 | 4.68 | |

OHCA, out-of-hospital cardiac arrest

anti-inflammatory and antiarrhythmic properties and can reduce the risk of developing atherosclerosis. However, our findings are in agreement with those of the prior studies, indicating that depression does not have an independent effect on OHCA risk. This suggest that while depression may be a risk factor for coronary heart disease and other cardiac events, it may not be a direct risk factor for OHCA.

Insomnia and depression, though separate disorders, have been shown to be highly interrelated in several previous studies. In a 2016 meta-analysis that analyzed the risk of depression in insomnia patients, the risk ratio of depression was 2.27 (95% CI: 1.89–2.71) among those with insomnia [34]. Additionally, symptoms of disturbed night-time sleep in individuals with depression have been suggested extensively in both clinical and epidemiological studies. In clinical studies, difficulty in initiating or maintaining sleep or both have been reported in about three-quarters of all depressive patients [35, 36]. In epidemiological studies examining insomnia and depression, sleep symptoms, including insomnia and hypersomnia, occurred in 50% to 60% in young adults [37]. In our study populations, 14.2% of insomnia patients had depression, while 52.7% of depression patients complained of insomnia. Regarding OHCA risk, insomnia and depression showed an interaction effect, and when insomnia was accompanied by depression, the OHCA risk was 2.33 times higher than when insomnia was not comorbid with depression, and this trend was maintained only in the young population under 65 years of age. A previous study on patients with coronary heart disease [38] reported that depression may be associated with mortality through negative health behaviors such as reduced physical activity.

Depression and insomnia are interconnected conditions that can negatively impact an individual's overall health and well-being. When both conditions are present, they may synergistic effect. For instance, depression is associated with neglecting health care needs, such as not seeking timely medical help, not adhering to prescribed treatments, and ignoring symptoms. This neglect can lead to worsening health outcomes and an increased risk of conditions such as OHCA. Insomnia, on the other hand, can contribute to the development or exacerbation of depression and further worsen an individual's health.

The relationship between insomnia and depression can create a vicious cycle, with each condition exacerbating the other. For example, chronic lack of restorative sleep due to insomnia may lead to the development or worsening of depressive symptoms. In turn, these depressive symptoms may disrupt sleep patterns and intensify insomnia. This continuous deterioration of both conditions not only heightens the risk of OHCA but also contributes to other adverse health outcomes. Moreover, insomnia and depression can intensify factors that increase OHCA risk, such as physiological changes, including increased inflammation, elevated stress hormones, and imbalances in the autonomic nervous system. The behavioral consequences of these conditions, like reduced physical activity, poor diet, and substance use, can further harm cardiovascular health.

A study was conducted in the UK on the incidence of insomnia in a wide range of patients of varying ages with depression. Overall, 83% of patients with depression had insomnia, which varied from 77% in the 16–24-year age group to 90% in the 55–64-year age group [39]. In our study, insomnia and depression showed a high trend in the elderly (those over 70 years of age); however, insomnia and depression did not increase the OHCA risk either alone or together in the elderly population.

In this study, neither insomnia nor depression alone increased the OHCA risk; however, insomnia with depression was associated with increased OHCA risk, and the effect of increased OHCA incidence was maintained only in the young population that did not consume insomnia medication. This may provide a theoretical basis for the need for early diagnosis and active treatment for insomnia patients with depression for lowering OHCA risk.

This study has several limitations. First, because histories of insomnia and depression were determined through self-report or through information obtained from the next of kin, rates of insomnia and depression may have been under- or over-estimated. Second, we did not investigate the treatment of depression in our registry questionnaire. There may be differences in the effect on OHCA risk depending on whether the disease is treated or not, which may have influenced the results of our study. Third, insomnia treatment other than medication could not be investigated in our registry. Finally, the study design was not a randomized controlled trial. There may have been a significant potential bias that was not controlled.

## Conclusions

A multicenter case-control study found that neither insomnia nor depression alone increased the risk of OHCA. However, an interaction analysis of insomnia and depression revealed that two conditions together increased the risk of OHCA in the young population without insomnia medication.

The study suggests a need for further research and strategies to reduce OHCA risk in general population, particularly in young population with insomnia and depression. Early and active medical intervention for insomnia patients with depression may contribute to lowering the risk of OHCA.

## Acknowledgments

We would like to acknowledge and thank to investigators from all 17 participating university hospitals of the phase II Cardiac Arrest Pursuit Trial, with Unique Registry and Epidemiologic Surveillance (CAPTURES-II): Sung Oh Hwang (Yonsei University Wonju Severance Christian Hospital), Sang Do Shin (Seoul National University Hospital), Mi Jin Lee (Kyungpook National University Hospital), Jong-Hak Park (Korea University Ansan Hospital), Su Jin Kim (Korea University Anam Hospital), Sung Bum Oh (Dankook University Hospital), Jonghwan Shin (Seoul National University Boramae Medical Center), Seung Min Park (Seoul National University Bundang Hospital), Min Seob Sim (Sungkyunkwan University Samsung Medical Center), Won Young Kim (Ulsan University Asan Medical Center), In-Cheol Park (Yonsei University Severance Hospital), Hyun Ho Ryu (Chonnam National University Hospital), Yeonho You (Chungnam National University Hospital), Sang-Chul Kim (Chungbuk National University Hospital), Ju Ok Park (Hallym University Dongtan Sacred Heart Hospital).

## Author Contributions

**Conceptualization:** Hyun Ho Ryu, Sung Wan Kim, Young Sun Ro.

**Data curation:** Hyun Ho Ryu, Sung Wan Kim, Kyoung Jun Song, Young Sun Ro.

**Formal analysis:** Jung Ho Lee.

**Investigation:** Jung Ho Lee, Young Sun Ro, Sung Oh Hwang.

**Methodology:** Kyoung Jun Song, Kyoung Chul Cha, Sung Oh Hwang.

**Project administration:** Young Sun Ro, Sung Oh Hwang.

**Resources:** Kyoung Jun Song.

**Software:** Eujene Jung, Hyun Ho Ryu, Young Sun Ro, Kyoung Chul Cha, Sung Oh Hwang.

**Supervision:** Eujene Jung, Hyun Ho Ryu, Sung Oh Hwang.

**Validation:** Eujene Jung, Hyun Ho Ryu.

**Visualization:** Eujene Jung, Hyun Ho Ryu, Kyoung Chul Cha.

**Writing – original draft:** Eujene Jung, Hyun Ho Ryu, Kyoung Chul Cha.

**Writing – review & editing:** Eujene Jung, Hyun Ho Ryu.

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
