## [Decision Letter · Decision Letter 0]

27 Jan 2023

PONE-D-22-19273Interaction Effects between Insomnia and Depression on Incidence of Out-of-Hospital Cardiac Arrest: Multi-center studyPLOS ONE

Dear Dr. Ho Ryu,

Thank you for submitting your manuscript to PLOS ONE. After careful consideration, we feel that it has merit but does not fully meet PLOS ONE’s publication criteria as it currently stands. Therefore, we invite you to submit a revised version of the manuscript that addresses the points raised during the review process.

We look forward to receiving your revised manuscript.

Kind regards,

Billy Morara Tsima, MD MSc

Academic Editor

PLOS ONE

Journal Requirements:

“This work was supported by the Korea Disease Control and Prevention Agency (Grant No: 2017NE3300600, 2017E3300601, 2019P330800).”

3. Please amend the manuscript submission data (via Edit Submission) to include author Sung Wan Kim, Jung Ho, Lee, sang do shin.

Additional Editor Comments (if provided):

Using the word 'incidence" in the context of this case-control study is confusing as it can imply the estimation of relative risk in epidemiological studies...which can not be calculated in a case-control study when the disease is not rare. However, I think the authors are not referring to incidence in this sense but rather "having an OHCA" or the incident of OHCA. It would be clearer if this is captured as "the risk of OHCA" only. Line 219-221, discusses incidence, which was not evaluated in this study.

There are a couple of grammatical errors; Line 135 (included IN this project), line 215 (those not taking vs those taking not).

Reviewers' comments:

Reviewer's Responses to Questions

**Comments to the Author**

1. Is the manuscript technically sound, and do the data support the conclusions?

Reviewer #1: Partly

Reviewer #2: Partly

2. Has the statistical analysis been performed appropriately and rigorously? 

Reviewer #1: I Don't Know

Reviewer #2: Yes

3. Have the authors made all data underlying the findings in their manuscript fully available?

Reviewer #1: No

Reviewer #2: Yes

4. Is the manuscript presented in an intelligible fashion and written in standard English?

Reviewer #1: Yes

Reviewer #2: Yes

5. Review Comments to the Author

Reviewer #1: 1. Introduction:

-The introduction seems well written. However, the definition of insomnia should be the one internationally accepted like from International Classification of Sleep Disorders, 3rd Edition.

2. Methods

-This section looks like it is a self-plagiarism, authors need to avoid that.

-The study design of this paper is a prospective multicenter case control study built from "The CAPTURES-II project was a prospective hospital-based patient cohort study conducted at 17 university hospitals from September 2017 to December 2020."Unfortunately there is no reference of this study to find out if it was an open or a close cohort.

-As this study calculate the OHCA incidence, it looks like it is a "density type case control study" (Incidence density or risk set sampling). So the cohort study from with it is coming from must be an open cohort.

-The matching technique used for the control group is appropriate as it always difficult to define the source population.

-Variables and measurements: the questions to elicit "insomnia" and "depression" are not scientifically acceptable. For instance, a participant may answer yes to a question “Have you ever suffered from insomnia or depression?" but this was not an insomnia or a depression. For instance, if one has poor sleep with no impact on daily activity, this is not an insomnia disorder. So there is a need to well define those concepts. Also, the authors did not define sleep apnea, heavy drinking vs. moderate drinking, vigorous vs. moderate physical activity...

3. Results

-The result section needs more flesh. There are a lot of information in table 1 and 2 that are not highlighted in the section.

4. Discussion:

-Chronic insomnia and depression are associated, this can be a confounder. I understand authors used regression model probably to control that. The anatomy of a discussion section should include major findings from the paper, similarities, and dissimilarities with previous studies, limitations, and recommendations. There is a need to expand more the discussion section.

5. Conclusion:

-The conclusion is just one paragraph.

Reviewer #2: The definition of insomnia and depression were not provided. Was difficulty sleeping for a day insomnia? was being sad for a week for example, depression? The cases and controls could understand it differently leading to misclassification. This will make controls also have the exposure of interest. Informed consent was waived. Was there any informed consent during collection of the primary data?

6. PLOS authors have the option to publish the peer review history of their article (what does this mean?). If published, this will include your full peer review and any attached files.

Reviewer #1: No

Reviewer #2: **Yes: **Goabaone Rankgoane-Pono

---

## [Author Response · Author response to Decision Letter 0]

16 May 2023

Author’s reply to reviewers' comments:

On behalf of authors, thank you for the very valuable comments by the reviewer on our paper. We have attempted to address every point commented on by the reviewer in the revised manuscript. While we believe that we have addressed all of the reviewer’s concerns, we would be more than pleased to write additional revisions if needed.

We highlighted all changes in red. Author’s answers or explanations are in blue.

Correspondent author

Hyun Ho Ryu, MD, PhD

 

Ref. No.: PONE-D-22-19273

Title: Interaction Effects between Insomnia and Depression on Incidence of Out-of-Hospital Cardiac Arrest: Multi-center study

Journal: PLOS ONE

Editor

Using the word 'incidence" in the context of this case-control study is confusing as it can imply the estimation of relative risk in epidemiological studies...which cannot be calculated in a case-control study when the disease is not rare. However, I think the authors are not referring to incidence in this sense but rather "having an OHCA" or the incident of OHCA. It would be clearer if this is captured as "the risk of OHCA" only. Line 219-221, discusses incidence, which was not evaluated in this study.

There are a couple of grammatical errors; Line 135 (included IN this project), line 215 (those not taking vs those taking not).

ANSWER: Thank you for the review. As you pointed out, ‘incidence’ was changed to ‘risk’ throughout the manuscript, and grammatical errors were revised.

 

Reviewers' comments

Reviewer's Responses to Questions

Comments to the Author

1. Is the manuscript technically sound, and do the data support the conclusions?

Reviewer #1: Partly

Reviewer #2: Partly

2. Has the statistical analysis been performed appropriately and rigorously?

Reviewer #1: I Don't Know

Reviewer #2: Yes

3. Have the authors made all data underlying the findings in their manuscript fully available?

Reviewer #1: No

Reviewer #2: Yes

4. Is the manuscript presented in an intelligible fashion and written in standard English?

Reviewer #1: Yes

Reviewer #2: Yes

5. Review Comments to the Author

Reviewer #1: 1. Introduction:

-The introduction seems well written. However, the definition of insomnia should be the one internationally accepted like from International Classification of Sleep Disorders, 3rd Edition.

ANSWER: Thank you for the review. We revised sentences and citation according to your advice.

(REVISION: Introduction): Insomnia, the most common sleep disorder, is characterized by difficulty initiating and maintaining sleep. It may also take the form of early-morning awakening in which the individual awakens several hours early and is unable to resume sleeping

2. Methods

-This section looks like it is a self-plagiarism, authors need to avoid that.

ANSWER: Thank you for the review. We revised methods section.

-The study design of this paper is a prospective multicenter case control study built from "The CAPTURES-II project was a prospective hospital-based patient cohort study conducted at 17 university hospitals from September 2017 to December 2020."Unfortunately there is no reference of this study to find out if it was an open or a close cohort.

ANSWER: Thank you for the review. Although, it was not specified that it was an open cohort, a study that explained the details of the cohort used in this study was cited.

-As this study calculate the OHCA incidence, it looks like it is a "density type case control study" (Incidence density or risk set sampling). So the cohort study from with it is coming from must be an open cohort.

ANSWER: Thank you for the review. Our cohort is an open cohort and reference was cited.

-The matching technique used for the control group is appropriate as it always difficult to define the source population.

ANSWER: Thank you for the review. The control group was randomly matched within the strata, including age, sex, and type of residence, with a 1:1 ratio of controls to cases.

-Variables and measurements: the questions to elicit "insomnia" and "depression" are not scientifically acceptable. For instance, a participant may answer yes to a question “Have you ever suffered from insomnia or depression?" but this was not an insomnia or a depression. For instance, if one has poor sleep with no impact on daily activity, this is not an insomnia disorder. So there is a need to well define those concepts. Also, the authors did not define sleep apnea, heavy drinking vs. moderate drinking, vigorous vs. moderate physical activity...

ANSWER: Thank you for the review. Insomnia and depression, the exposures of our study, were diagnosed history, and the Method section was revised. Also, the process of defining other variables was also added to the manuscript in detail. 

(REVISION: Methods-Variables and measurements): Insomnia history was defined as a positive case of answering “yes” to the question “Have you ever been diagnosed with insomnia?” Depression history was defined in the same way.

(REVISION: Methods-Variables and measurements): Whether or not comorbidities and sleep-related history were diagnosed by a doctor was investigated. Alcohol drinking was defined as heavy alcohol drinking when men drank more than 15 drinks per week and women drank more than 8 drinks per week and physical activity was defined as vigorous when exercising to the extent of sweating at least once a week, and moderate when exercising less than once a week.

3. Results

-The result section needs more flesh. There are a lot of information in table 1 and 2 that are not highlighted in the section.

ANSWER: Thank you for the review. We revised Results section according to your advice.

(REVISION: Results-Main results): A total of 1014 OHCA cases and 1014 community-based controls were included in this analysis. 

The characteristics of the OHCA cases and community-based controls are shown in Table 1. And, the characteristics according to the history of insomnia and depression are reported Table 2 and Table 3. 

The results of conditional logistic regression models, including AORs (95% CIs), for OHCA with history of insomnia and depression are shown in Table 4. Insomnia was not associated with increased OHCA risk (AOR [95% CI]: 0.95 [0.64—1.40]). Depression also was not a risk factor of OHCA (0.86 [0.67—1.10]).

4. Discussion:

-Chronic insomnia and depression are associated, this can be a confounder. I understand authors used regression model probably to control that. The anatomy of a discussion section should include major findings from the paper, similarities, and dissimilarities with previous studies, limitations, and recommendations. There is a need to expand more the discussion section.

ANSWER: Thank you for the review. We revised discussion section according to your advice. 

5. Conclusion:

-The conclusion is just one paragraph.

ANSWER: Thank you for the review. The overall content of study was added to the Conclusion.

(REVISION: Conclusions): A multicenter case-control study found that neither insomnia nor depression alone increased the risk of OHCA. However, an interaction analysis of insomnia and depression revealed that two conditions together increased the risk of OHCA in the young population without insomnia medication. 

The study suggests a need for further research and strategies to reduce OHCA risk in general population, particularly in young population with insomnia and depression. Early and active medical intervention for insomnia patients with depression may contribute to lowering the risk of OHCA. 

Reviewer #2: The definition of insomnia and depression were not provided. Was difficulty sleeping for a day insomnia? was being sad for a week for example, depression? The cases and controls could understand it differently leading to misclassification. This will make controls also have the exposure of interest. Informed consent was waived. Was there any informed consent during collection of the primary data?

ANSWER: Thank you for the review. Insomnia history was defined as a positive case of answering “yes” to the question “Have you ever been diagnosed with insomnia?” Depression history was defined in the same way. All study participants or their proxies provided a written informed consent prior to their participation in the study.

(REVISION: Methods-Study design, setting, and data sources): The study was approved by the ethics committees in all participating institutions. In compliance with the Declaration of Helsinki, all participants, or their proxies, provided a written informed consent prior to their participation in the study (35000357). This study is registered at ClinicalTrials.gov (NCT03700203).

(REVISION: Methods-Variables and measurements): Insomnia history was defined as a positive case of answering “yes” to the question “Have you ever been diagnosed with insomnia?” Depression history was defined in the same way.

6. PLOS authors have the option to publish the peer review history of their article (what does this mean?). If published, this will include your full peer review and any attached files.

Do you want your identity to be public for this peer review? For information about this choice, including consent withdrawal, please see our Privacy Policy.

Reviewer #1: No

Reviewer #2: Yes: Goabaone Rankgoane-Pono

---

## [Decision Letter · Decision Letter 1]

15 Jun 2023

Interaction Effects between Insomnia and Depression on Incidence of Out-of-Hospital Cardiac Arrest: Multi-center study

PONE-D-22-19273R1

Dear Dr. Hyun Ho Ryu,

We’re pleased to inform you that your manuscript has been judged scientifically suitable for publication and will be formally accepted for publication once it meets all outstanding technical requirements.

Kind regards,

Billy Morara Tsima, MD MSc

Academic Editor

PLOS ONE

Additional Editor Comments (optional):

Reviewers' comments:

Reviewer's Responses to Questions

**Comments to the Author**

1. If the authors have adequately addressed your comments raised in a previous round of review and you feel that this manuscript is now acceptable for publication, you may indicate that here to bypass the “Comments to the Author” section, enter your conflict of interest statement in the “Confidential to Editor” section, and submit your "Accept" recommendation.

Reviewer #1: All comments have been addressed

Reviewer #2: All comments have been addressed

2. Is the manuscript technically sound, and do the data support the conclusions?

Reviewer #1: Yes

Reviewer #2: Yes

3. Has the statistical analysis been performed appropriately and rigorously? 

Reviewer #1: Yes

Reviewer #2: Yes

4. Have the authors made all data underlying the findings in their manuscript fully available?

Reviewer #1: Yes

Reviewer #2: Yes

5. Is the manuscript presented in an intelligible fashion and written in standard English?

Reviewer #1: Yes

Reviewer #2: Yes

6. Review Comments to the Author

Reviewer #1: 1. Abstract: Authors should review the abstract as it looks like they did not make any change after reviewing their manuscript.

2. Introduction: Well done introduction. However, it could be better to add a sentence or paragraph on findings from previous study on the association of insomnia +/-depression AND OHCA. Or state whether such studies are scarce. Also Shift the "purpose" sentence should come before the hypothesis sentence (lines 135- 136 to come before 133-134).

3. Methods: I suggest moving lines 161-164 to ethical considerations section.

4. General comments: Proofread the English, use past tense in methods section. For instance: line 143 "The CAPTURES project is a prospective, multi-center, case-control study, which aims to identify risk..."

Reviewer #2: The authors have addressed the comments well. The manuscript is scientifically sound. The research methods have been clearly described.

7. PLOS authors have the option to publish the peer review history of their article (what does this mean?). If published, this will include your full peer review and any attached files.

Reviewer #1: **Yes: **Stephane Tshitenge

Reviewer #2: **Yes: **Goabaone Rankgoane-Pono

---

## [Editor Report · Acceptance letter]

10 Aug 2023

PONE-D-22-19273R1 

Interaction Effects between Insomnia and Depression on Risk of Out-of-Hospital Cardiac Arrest: Multi-center study 

Dear Dr. Ryu:

I'm pleased to inform you that your manuscript has been deemed suitable for publication in PLOS ONE. Congratulations! Your manuscript is now with our production department. 

Kind regards, 

on behalf of

Dr. Billy Morara Tsima 

Academic Editor

PLOS ONE